

# Genome-wide identification and expression analysis of diacylglycerol acyltransferase genes in soybean (*Glycine max*)

Shihui Zhao[*], Fan Yan[*], Yajing Liu, Monan Sun, Ying Wang, Jingwen Li, Xinsheng Zhang, Xuguang Yang and Qingyu Wang

College of Plant Science, Jilin University, Changchun, China
[*] These authors contributed equally to this work.

## ABSTRACT

**Background.** Soybean (*Glycine max*) is a major protein and vegetable oil source. In plants, diacylglycerol acyltransferase (DGAT) can exert strong flux control, which is rate-limiting for triacylglycerol biosynthesis in seed oil formation.

**Methods.** Here, we identified soybean *DGAT* genes via a bioinformatics method, thereby laying a solid foundation for further research on their function. Based on our bioinformatics analyses, including gene structure, protein domain characteristics, and phylogenetic analysis, 26 *DGAT* putative gene family members unevenly distributed on 12 of the 20 soybean chromosomes were identified and divided into the following four groups: *DGAT1*, *DGAT2*, *WS/DGAT*, and cytoplasmic *DGAT*.

**Results.** The Ka/Ks ratio of most of these genes indicated a significant positive selection pressure. *DGAT* genes exhibited characteristic expression patterns in soybean tissues. The differences in the structure and expression of soybean *DGAT* genes revealed the diversity of their functions and the complexity of soybean fatty acid metabolism. Our findings provide important information for research on the fatty acid metabolism pathway in soybean. Furthermore, our results will help identify candidate genes for potential fatty acid-profile modifications to improve soybean seed oil content.

**Conclusions.** This is the first time that *in silico* studies have been used to report the genomic and proteomic characteristics of DGAT in soybean and the effect of its specific expression on organs, age, and stages.

Corresponding authors
Xuguang Yang, xgyang@jlu.edu.cn
Qingyu Wang, qywang@jlu.edu.cn

## INTRODUCTION

Soybean (*Glycine max*) is a major protein and vegetable oil source for humans. Lipids are primarily stored in the soybean seed as triacylglycerols (TAGs), which comprise three C16–C18 non-esterified fatty acids esterified to a glycerol molecule (*Mekhedov, De Ilárduya & Ohlrogge, 2000*). Soybean lipids are also stored in the leaf blade and stem *via* the formation of wax esters comprising aliphatic alcohols and acids, with both moieties being typically long-chain (C16 and C18) or very-long-chain (C20–C34 or longer) carbon

structures (*Yeats & Rose, 2013*). In plants, diacylglycerol acyltransferase (DGAT) can exert strong flux control in TAG biosynthesis for seed oil formation (*Barthole et al., 2012*; *Chen et al., 2022*). Specifically, DGAT catalyzes the formation of an ester linkage between a fatty acyl-CoA and the free hydroxyl group of diacylglycerol to form TAGs (*Liu et al., 2012*). According to their structure and cellular or subcellular localization, DGATs are divided into four types: DGAT1 (*Cases et al., 1998*; *Lardizabal et al., 2008*), DGAT2 (*Cases et al., 2001*; *Lardizabal et al., 2001*), WS/DGAT (*Kalscheuer & Steinbüchel, 2003*), and cytoplasmic DGAT (cytoDGAT) (*Saha et al., 2006*). DGAT1 belongs to the acyl-CoA cholesterol acyltransferase family, whereas DGAT2 belongs to the DGAT2 superfamily; further, DGAT1 and DGAT2 primarily bind to the endoplasmic reticulum membrane and are microsomal enzymes. WS/DGATs, which are bifunctional enzymes that play a key role in wax-ester synthesis (*Cases et al., 1998*; *Lardizabal et al., 2008*), and cytoDGAT, recently discovered, have been relatively less studied (*Kalscheuer & Steinbüchel, 2003*; *Saha et al., 2006*). The DGAT1 protein subfamily members are expressed in both animals and plants. After cloning of the first *DGAT1* gene in mice (*Cases et al., 2001*), *Hobbs, Lu & Hills (1999)* first cloned *DGAT1* in plants, specifically in Arabidopsis, which has only one copy of the gene. Subsequently, *DGAT1* has been cloned in other plant species (*Xu et al., 2008*; *Zheng et al., 2008*). *DGAT1* (50–60 kDa and >500 amino acid residues) is expressed in flowers and developing and germinating seeds. The three-dimensional structure of the protein contains 9–10 hypothetical transmembrane regions and a hydrophilic N terminus (*Nykiforuk et al., 2002*; *He et al., 2004a*; *He et al., 2004b*) that faces the cytoplasmic surface of the endoplasmic reticulum membrane (*Shockey et al., 2006*). Structural differences in *DGAT1* among species primarily exist in the N terminus (*Zou et al., 1999*) and may be related to the selectivity of DGAT1 to the acyl-coenzyme A substrate in different plant species.

The *DGAT2* subfamily members are expressed in animals, plants, and yeast (*Kroon et al., 2006*; *Oakes et al., 2011*). DGAT2 (30–40 kDa, >300 amino acid residues) from *Mortierella ramanniana* liposomes was first cloned into other plant species by *Routaboul et al. (1999)*. In turn, *Kalscheuer & Steinbüchel (2003)* identified a gene encoding a bifunctional enzyme with both wax-ester synthesis and *DGAT* functions in *Acinetobacter calcoaceticus*. This *WS/DGAT* enzyme (51.8 kDa, 458 amino acid residues) is unrelated to the known wax-ester synthases *DGAT1* and *DGAT2*.

*CytoDGAT* is a novel *DGAT* gene that was cloned from immature peanut seeds (*Hobbs, Lu & Hills, 1999*). The gene encodes a 38 kDa protein containing 345 amino acid residues. The protein is localized in the cytoplasm of cotyledonary cells and does not contain any hypothetical transmembrane region. Its amino acid sequence is 13% homologous to that of *WS/DGAT* and less than 10% homologous to that of the *DGAT1* and *DGAT2* family members. Therefore, the *DGAT* genes in soybean were analyzed in this study using systematic identification and characterized through phylogenetic, structural diversification, and expression profile analyses.

There are many studies regarding the *DGAT* gene in soybean, most of them focusing on its function. *Wang et al. (2006)* cloned the *DGATT1* and *DGATT2* genes from wild-type and cultivated soybean using the RACE method. *Li et al. (2013)* cloned the *DGAT1A* and

*DGAT1B* genes from JACK varieties. However, *DGAT* in standard soybean varieties has not yet been fully reported. The results of this study will aid in understanding the evolution and function of the *GmDGAT* gene family in soybean as well as the genomic architectures and biological functions of each subfamily. Our study could provide a comprehensive account of the *GmDGAT* gene family in soybean and also serve as a starting point for further elucidation of their roles in soybean seed oil synthesis and accumulation as well as plant growth and development.

## MATERIALS & METHODS

### Sequence retrieval and data analysis

To identify all putative DGAT proteins in soybean, *G. max* genome data were downloaded from the latest NCBI database (http://www.ncbi.nlm.nih.gov/). Data were collected as previously described in *Zhao (2018)*. According to Arabidopsis DGAT1 and DGAT2 proteins in the Pfam database (http://pfam.xfam.org/), DGAT proteins should have a PF03982 domain and WS/DGAT should have Pfam: PF03007 and PF06974 domains. These protein domains were used as queries in Hmmer3.0 with the *E*-value set at 1E-005. Based on the amino acid sequence in peanuts, the DGAT3 (cytoDGAT) protein was searched and screened *via* BLAST in the soybean database. The exon/intron organization for *G. max DGAT* genes was defined by loading its coding sequence (CDS) and the corresponding genomic sequence into Phytozome13 (Phytozome). Molecular weights and isoelectric points of candidate GmDGAT proteins were analyzed using ExPASy (http://web.expasy.org/protparam/). Predicting subcellular localization of *DGAT* genes in soybean was performed using Cell-PLoc 2.0 (*Chou & Shen, 2010*) (http://www.csbio.sjtu.edu.cn/bioinf/plant-multi/), DeepLoc (*Armenteros et al., 2017*), and WoLF pSORT (www.genscript.com/wolf-psort.html) (*Horton et al., 2007*).

### Conserved domain, motif, CDS, and phylogenetic analyses

To identify the conserved domains, motifs, and CDS in the GmDGAT proteins, we used the Pfam (http://pfam.xfam.org/) and MEME (https://meme-suite.org/meme/) databases and Gene Structure Display Server 2.0 (GSDS2.0, http://gsds.cbi.pku.edu.cn/), respectively. Using MEGA X (https://www.megasoftware.net/), we constructed a phylogenetic tree with the Jones–Taylor–Thornton model; bootstrap tests were performed using 1,000 replicates for statistical reliability.

### Cis-element, chromosomal localization, and collinearity analyses of GmDGAT genes

The 2-kb region upstream of the start codon of the *DGAT* genes served as the promoter sequences that were submitted to PlantCARE (http://bioinformatics.psb.ugent.be/webtools/plantcare/html/) for predicting cis-acting elements in the promoter region. The chromosomal location of *DGAT* genes was obtained from Phytozome (https://phytozome-next.jgi.doe.gov/). MCScanX was used to determine the syntenic relationships among *DGAT* gene family members. The TBTools software was used to describe the distribution of the identified *cis*-elements, chromosome location map, and collinearity analysis results.

## Estimating Ka/Ks ratios for duplicated gene pairs

In order to understand the mechanisms of DNA sequence evolution, reconstruct phylogenic trees, and identify protein coding exons, nonsynonymous (amino acid replacing) and synonymous (silent) substitution rates between protein coding sequences were used to determine whether there was selection pressure acting on this protein coding gene (*Yang & Nielsen, 2000*; *Nekrutenko, 2002*). Nonsynonymous (amino acid substitution) and synonymous (silent) substitutions are called Ka and Ks, respectively. Ka reflects nonsynonymous substitutions per nonsynonymous site, and Ks reflects synonymous substitutions per synonymous site. The Ka/Ks ratio (denoted as $\omega$) is widely used as an estimator of selective strength for DNA sequence evolution, with $\omega > 1$ indicating positive selection, $\omega < 1$ indicating purifying (negative) selection, and $\omega$ close to 1 indicating neutral mutation (*Chattopadhyay et al., 1998*). Data are processed according to the protocol described by *Aylward (2018)*, calculated the Ka/Ks ratios of *GmDGATs*.

## GmDGAT expression pattern analysis

Based on the soybase database's (https://soybase.org/soyseq/) published sequencing results of the soybean whole transcriptome, we screened relevant data and analyzed *GmDGAT* expression in different tissues from the expression data available for 51,529 soybean genes. We used Heatmap in the R software (version 2.15.2) to construct a heat map.

## RT-PCR analysis of GmDGAT genes

The soybean variety Williams 82 (w82), used in this experiment, was provided by the Institute of Plant Sciences of Jilin University and planted at its variety trial station (43°55′22.63″N, 125°16′3216′33.00″E). Roots, stems, leaves, flowers, and seeds of soybean plants were collected 10, 20, 30, 40, 50, and 60 days after flowering (DAF) for subsequent analysis. Tissue samples were frozen in liquid nitrogen and stored in refrigerators at −80 °C. The E.Z.N.A.® plant RNA Kit (Omega Biotechnology, Norcross, GA, USA) was used for extracting total RNA from all collected samples according to the manufacturer's instructions. The ReverTra Ace PCR RT Master Mix with gDNA Remover (TOYOBO, Japan) was used to reverse-transcribe RNA into first-strand cDNA. Subsequently, RT-PCR was conducted using QuantStudio™ 5 Real-Time PCR software (Foster City, CA, USA) and KOD SYBR qPCR Mix (TOYOBO) following the protocol proposed by the manufacturer. Three biological and three technical replicates were used for each treatment in the RT-PCR analysis. Soybean *GmActin* was used as a reference gene. The $2^{-\Delta\Delta C(t)}$ method was used to determine the relative gene expression levels of 10 randomly selected *GmDGAT* genes. The primers used for RT-PCR are listed in Table S1.

# RESULTS

## Identification of DGAT proteins in soybean (G. max)

A total of 26 DGAT putative proteins were identified: three proteins containing the MBOAT (PF03062) protein domain, 10 proteins containing the DGAT (PF03982) protein domain, 12 proteins containing the WS/DGAT (PF03007 and PF06974) protein domains (Fig. 1Bi), which were identified in soybean in the latest version of the *G. max* genome assembly

(Wm82.a4.v1). Using the peanut cytoDGAT sequence as a reference query, the sequence alignment program BLASTP identified one cytoDGAT protein (Glyma.13G118300) in soybean.

Among the identified DGAT proteins, three were assigned to the DGAT1 subfamily because they contained the membrane-bound o-acyltransferase (MBOAT) domain. In turn, 10 genes were assigned to the DGAT2 subfamily as they contained the monoacylglycerol acyltransferase domain; these genes were further classified into DGAT2a and DGAT2b according to the similarity of their encoded amino acid sequences and motifs. Meanwhile, the WS/DGAT subfamily identified 12 genes containing the acyl-CoA wax-alcohol acyltransferase domain; these genes were further classified into WS/DGATa, WS/DGATb, and WS/DGATc. CytoDGAT was found to contain only one gene.

We evaluated the basic properties of these DGAT proteins, including gene name, gene ID, CDS length, isoelectric point, molecular weight, and number of exons (Table 1). The amino acid (aa) length of DGATs varied from 249 to 704 aa and the isoelectric point (pI) ranged from 6.06 to 10.03. There were similarities in basic gene and protein properties within as well as among groups.

The subcellular localization prediction results showed that *GmDGATs* were centrally localized in the chloroplast, followed by the cell membrane, cytoplasm, nucleus, mitochondria, peroxisome, and golgi apparatus (Table 1). *GmDGAT2a5*, *GmDGAT2b3*, *GmDGAT2b4*, and *GmDGAT2b5* failed to predict their subcellular location.

## Conserved domains and phylogenetic analysis

Multiple alignment analyses were conducted to determine the classification and evolutionary relationships of *DGAT* genes in soybean, and an unrooted maximum-likelihood phylogenetic tree was constructed using the MEGA 5.10 software with the amino acid sequences of the 26 soybean DGAT proteins. Based on the phylogenetic tree (Fig. 1Ai), soybean *DGAT* genes were divided into four subfamilies: *DGAT1*, *DGAT2*, *WS/DGAT*, and *cytoDGAT*. Three DGAT proteins were included in the DGAT1 protein subfamily. The members of the DGAT2 subfamily were divided into two subgroups, each possessing five members of DGAT2a and DGAT2b. In turn, the WS/DGAT subfamily members were divided into three subgroups that possessed 7, 3, and 2 members from WS/DGATa, WS/DGATb, and WS/DGATc, respectively.

To understand the structural diversity of DGAT, the conserved domains (Fig. 1Bi) were analyzed with Pfam. All DGAT1 subfamily members contained the MBOAT domain, and all DGAT2 subfamily members contained the DGAT domain. Furthermore, the DGAT2b family included the Abhydrolase_6 domain, DGAT2b1 contained the Hydrolase_4 domain, and DGAT2b2 and DGAT2b5 contained the Abhydrolase_1 domain. Additionally, the WS/DGAT subfamily contained the WES_Acyltransf and WS_DGAT_C domains (Fig. 1Bii). In contrast, cytoDGAT did not contain any of the conserved domains but contained a unique 2Fe-2S_Thioredoxin domain. The conserved motifs of the *DGAT* genes were analyzed using the MEME motif search tool, which revealed 15 significant (e-value <e-100) conserved motifs (Fig. 1Aii). Different subgroups contained different motifs, including numbers and composition; however, the same group in the phylogenetic tree had the same

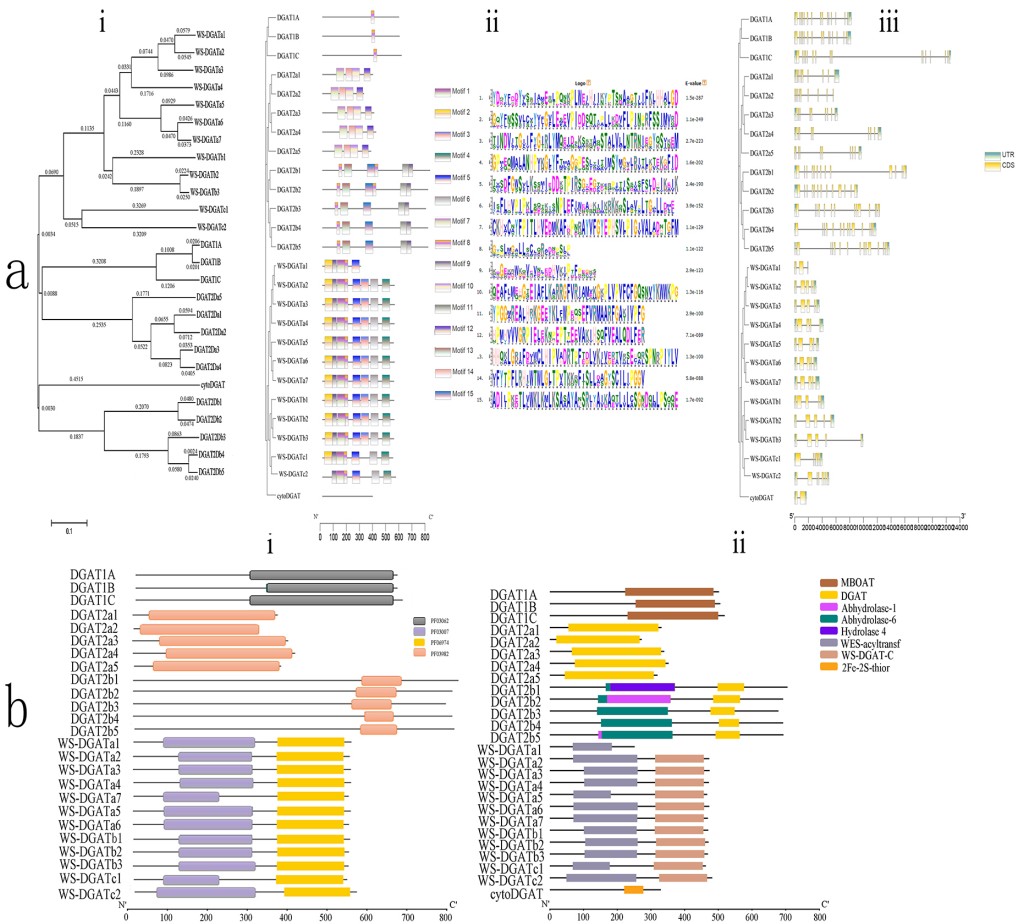

**Figure 1** **Conserved domain, motif, CDS, and phylogenetic analyses.** (A) Phylogenetic relationships, motifs, and coding sequence (CDS) of soybean diacylglycerol acyltransferase (DGAT) genes in soybean. (i) Phylogenetic tree based on the full-length sequence of soybean DGAT proteins built using MEGAX software. (ii) Fifteen predicted motifs are represented by boxes of different colors, and their sizes are indicated by the scale on the right. (iii) Exon/intron structure of *GmDGAT* genes. Green boxes represent untranslated 5′- and 3′-regions, yellow boxes represent exons, and black lines represent introns. (B) Organization of conserved domains in the identified DGAT proteins. (i) The signature domains of GmDGAT protein analyzed with Pfam. (ii) The conserved protein domain of GmDGAT protein.

conserved motif structures. Motif8 existed in all DGAT1 subfamily members. Subgroup DGAT2a had motif7, motif10, motif12, and motif14, and subgroup DGAT2b had motif11, motif13, and motif15; the WS/DGAT subfamily contained motif8 and motif9. In contrast, the cytoDGAT subfamily had none of these motifs.

The exon/intron organization (Fig. 1Aiii) showed similar numbers of introns and exons in one subgroup. All members of *DGAT1* had 16 exons. The *DGAT2a* members had 9 exons, and *DGAT2b* subgroup members had 13–14 exons. The number of introns in the *WS/DGAT* subfamily varied from three to seven. *WS/DGATa* and *WS/DGATb* members mostly contained five exons, *WS/DGATa1* contained three exons, *WS/DGATc* had six and seven exons, and *CtyoDGAT* had two exons.

**Table 1** Characteristics of soybean diacylglycerol acyltransferase genes and Proteins.

| Gene name | Gene ID | cDNA length (bp) | pI | MW | Number of amino acid residues | Predicting subcellular localization |
|---|---|---|---|---|---|---|
| GmDGAT1A | Glyma.13G106100 | 1503 | 8.89 | 57563.72 | 500 | Cell membrane. Chloroplast. Golgi apparatus. |
| GmDGAT1B | Glyma.17G053300 | 1515 | 8.89 | 57999.11 | 504 | Cell membrane. |
| GmDGAT1C | Glyma.09G065300 | 1554 | 8.93 | 59286.5 | 517 | Cell membrane. |
| GmDGAT2a1 | Glyma.01G156000 | 990 | 9.7 | 36884.63 | 329 | Cell membrane. Chloroplast. |
| GmDGAT2a2 | Glyma.11G088800 | 813 | 10.03 | 30762.67 | 270 | Chloroplast. |
| GmDGAT2a3 | Glyma.09G195400 | 1014 | 9.62 | 38202.02 | 337 | Chloroplast. |
| GmDGAT2a4 | Glyma.16G115700 | 1053 | 9.68 | 39591.78 | 350 | Cell membrane. Chloroplast. |
| GmDGAT2a5 | Glyma.16G115800 | 954 | 9.26 | 36002.63 | 317 | – |
| GmDGAT2b1 | Glyma.03G243700 | 2115 | 7.02 | 78293.74 | 704 | Chloroplast. |
| GmDGAT2b2 | Glyma.19G241200 | 2076 | 6.06 | 76460.35 | 691 | Chloroplast. |
| GmDGAT2b3 | Glyma.16G051300 | 2034 | 9 | 76186.14 | 677 | – |
| GmDGAT2b4 | Glyma.16G051200 | 2076 | 6.99 | 76886.72 | 691 | – |
| GmDGAT2b5 | Glyma.19G099400 | 2079 | 8 | 77309.55 | 692 | – |
| GmWS/DGATa1 | Glyma.12G114400 | 750 | 6.3 | 28375.42 | 249 | Cytoplasm. |
| GmWS/DGATa2 | Glyma.06G291700 | 1416 | 9.04 | 53801.41 | 471 | Chloroplast. |
| GmWS/DGATa3 | Glyma.06G291300 | 1419 | 9.48 | 53807.73 | 472 | Chloroplast. |
| GmWS/DGATa4 | Glyma.12G114900 | 1413 | 7.66 | 52957.1 | 470 | Chloroplast. |
| GmWS/DGATa5 | Glyma.18G258100 | 1398 | 6.68 | 52264.01 | 465 | Cell membrane. Chloroplast. |
| GmWS/DGATa6 | Glyma.09G239600 | 1416 | 6.47 | 52783.67 | 471 | Chloroplast. |
| GmWS/DGATa7 | Glyma.18G257900 | 1404 | 8.62 | 52417.41 | 467 | Chloroplast. |
| GmWS/DGATb1 | Glyma.13G295900 | 1407 | 9.01 | 52574.35 | 468 | Cell membrane. Cytoplasm. |
| GmWS/DGATb2 | Glyma.19G046000 | 1410 | 8.77 | 52897.1 | 469 | Chloroplast. |
| GmWS/DGATb3 | Glyma.13G046600 | 1404 | 8.18 | 52713.85 | 467 | Chloroplast. |
| GmWS/DGATc1 | Glyma.07G000300 | 1386 | 8.37 | 51291.55 | 461 | Chloroplast. Nucleus. Peroxisome. |
| GmWS/DGATc2 | Glyma.09G196400 | 1443 | 9.35 | 53849.94 | 480 | Chloroplast. |
| GmcytoDGAT | Glyma.13G118300 | 984 | 8.52 | 34732.81 | 327 | Chloroplast. Cytoplasm. Mitochondrion. |

**Notes.**
DGAT, diacylglycerol acyltransferase; MW, molecular weight; pI, isoelectric point.

## Chromosomal localization and collinearity analysis

The chromosomal location of the *DGAT* genes was plotted according to the physical positions of the *DGAT* genes on soybean chromosomes (Fig. 2) (*Kim et al., 2017*). The 26 *DGAT* genes were unevenly distributed on 12 of the 20 chromosomes in soybean. Chromosomes 9, 13, and 16 had four genes; chromosome 19 had three genes; chromosomes 6, 12, and 18 had two genes; and chromosomes 3, 7, 11, and 17 had only one gene. To understand the evolution and amplification of DGAT genes in the soybean genome, we performed collinearity analysis. All segmentally duplicated gene pairs belonged to the same phylogenetic subfamilies, likely because, as an ancient tetraploid, soybean shows many chromosomal rearrangements, high genomic duplication, and multiple gene copies (*Schmutz et al., 2010*). DGAT1a, DGAT1b, and DGAT1c, located on chromosomes 9, 13, and 17, respectively, were collinear.

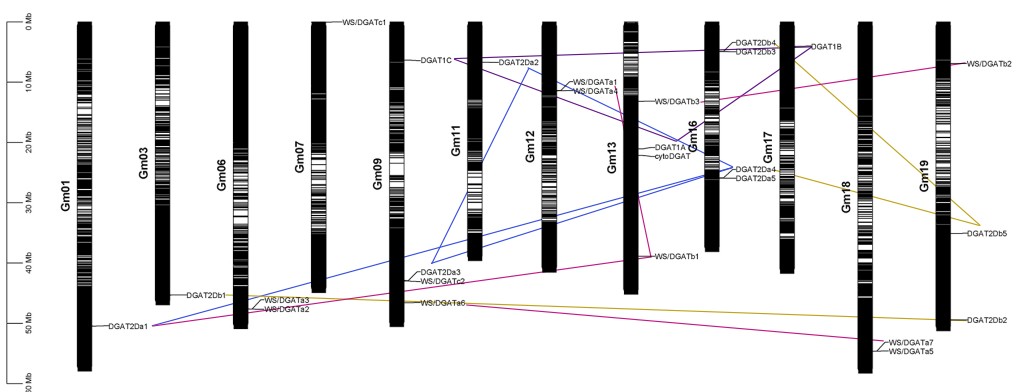

**Figure 2  Chromosomal localization, and collinearity analyses of GmDGAT genes.** Distribution of 26 *DGAT* genes on soybean (*Glycine max* L.) chromosomes. Collinearity analysis of soybean *DGAT* genes. Red, blue, green, and brown lines indicate *DGAT1*, *DGAT2a*, *DGAT2b*, and *WS/DGAT* genes, respectively.

Synteny and collinearity is used to clarify angiosperm evolution and reveal patterns of differential gene loss after genome duplication and different gene retention associated with evolution of some morphological complexity (*Tang et al., 2008*). Furthermore, *DGAT2a1*, *DGAT2a2*, *DGAT2a3*, and *DGAT2a4* showed collinearity, whereas *DGAT2a5* was only collinear with *DGAT2a4*. In turn, *DGAT2b1* and *DGAT2b2* were collinear, and *DGAT2b* 4 was collinear with *DGAT2b* 3 and *DGAT2b* 5. Furthermore, *WS/DGAT* a6 and *WS/DGAT* a7 were collinear, and *WS/DGAT* b1 was collinear with *WS/DGAT* a1 and *WS/DGAT* a3. The results of collinearity analysis show that DGAT subfamilies evolved independently. Meanwhile, gene duplication and orphan genes occur in subfamilies. A collinear relationship was also found between *WS/DGAT* b2 and *WS/DGAT* b3. Due to the chromosome doubling event in soybean evolution, chromosome mapping and collinearity can help us analyze which genes are generated by chromosome reduplication and which genes are generated by DNA fragment doubling. Genes with collinearity may have come from the same original gene during gene evolution.

## Selective pressure on DGAT genes

The replacement times of non-synonymous replacement sites (Ka) to the replacement times of synonymous site (Ks). The Ka/Ks ratio is usually used to insight the pressure on the selective evolution of each gene in the whole gene family. In this study, 35 paralogous gene pairs were determined among the *DGAT* genes using the Clustal W software (Table 2). Based on 6.161029 synonymous mutations for soybean per synonymous site per year, the Ka, Ks, and Ka/Ks values varied from 0.165 to 0.539, 0.138 to 4.24, and 0.164 to 4.412 Mya, respectively. Most gene pairs showed an estimated Ka/Ks >1, suggesting they are under significant positive selection pressure. Moreover, members of duplicated gene pairs seemingly indicated positive selection. For pairs *DGAT2a3–DGAT2b2*, *WS/DGATa6–WS/DGATc2*, *DGAT2a4–DGAT2b5*, *DGAT2a3–DGAT2b5*, *WS/DGATa3–WS/DGATc2*, *WS/DGATb2–WS/DGATc2*, and *WS/DGATa5–WS/DGATa6*, the Ka/Ks ratio was estimated

**Table 2** **Divergence and duplication of *DGAT* gene pairs in soybean.**

| Duplicated pair | | Ka | Ks | Ka/Ks | P-Value | Divergence Time | Selection pressure |
|---|---|---|---|---|---|---|---|
| DGAT1A | DGAT1B | 0.17 | 0.16 | 1.04 | 0.75 | 0.16 | |
| DGAT2a1 | DGAT2a4 | 2.89 | 2.55 | 1.13 | 0.01 | 2.81 | |
| DGAT2a1 | DGAT2b1 | 3.02 | 1.25 | 2.43 | 0 | 2.55 | |
| DGAT2a1 | DGAT2a3 | 5 | 1.09 | 4.59 | 0 | 3.91 | |
| DGAT2a2 | DGAT2b1 | 3.19 | 2.48 | 1.28 | 0.67 | 3.02 | |
| DGAT2a3 | DGAT2b2 | 2.09 | 4.15 | 0.5 | 0 | 2.53 | purifying |
| DGAT2a3 | DGAT2b5 | 2.74 | 4.08 | 0.67 | 0.15 | 3.04 | purifying |
| DGAT2a3 | DGAT2a4 | 2.85 | 1.16 | 2.45 | 0 | 2.43 | |
| DGAT2a3 | DGAT2b3 | 5.02 | 1.66 | 3.03 | 0 | 4.15 | |
| DGAT2a4 | DGAT2b5 | 2.43 | 4.11 | 0.59 | 0.19 | 2.81 | purifying |
| DGAT2a4 | DGAT2b3 | 2.7 | 2.57 | 1.05 | 0.87 | 2.67 | |
| DGAT2a4 | DGAT2b2 | 5.07 | 2.18 | 2.32 | 0.02 | 4.31 | |
| DGAT2a5 | DGAT2b1 | 2.29 | 1.44 | 1.6 | 0.03 | 2.1 | |
| DGAT2b1 | DGAT2b2 | 0.17 | 0.14 | 1.21 | 0.15 | 0.16 | |
| DGAT2b1 | DGAT2b5 | 2.49 | 1.55 | 1.6 | 0.01 | 2.25 | |
| DGAT2b1 | DGAT2b3 | 5.4 | 1.81 | 2.99 | 0 | 4.41 | |
| WS/DGATa1 | WS/DGATa4 | 2.5 | 2.28 | 1.1 | 0.75 | 2.45 | |
| WS/DGATa1 | WS/DGATa5 | 3.59 | 2.1 | 1.71 | 0.01 | 3.24 | |
| WS/DGATa2 | WS/DGATa5 | 5.13 | 1.35 | 3.8 | 0 | 4.11 | |
| WS/DGATa2 | WS/DGATa4 | 5.2 | 1.26 | 4.13 | 0 | 4.13 | |
| WS/DGATa3 | WS/DGATc2 | 2.16 | 3.05 | 0.71 | 0.36 | 2.35 | purifying |
| WS/DGATa3 | WS/DGATb1 | 3.14 | 2.06 | 1.52 | 0.83 | 2.91 | |
| WS/DGATa3 | WS/DGATa5 | 3.32 | 1.71 | 1.94 | 0 | 2.92 | |
| WS/DGATa4 | WS/DGATa7 | 0.56 | 0.56 | 1 | 0.63 | 0.56 | |
| WS/DGATa4 | WS/DGATc1 | 5.18 | 1.42 | 3.65 | 0 | 4.15 | |
| WS/DGATa5 | WS/DGATa6 | 2.51 | 2.92 | 0.86 | 0 | 2.6 | purifying |
| WS/DGATa5 | WS/DGATc1 | 3.03 | 1.67 | 1.81 | 0.02 | 2.69 | |
| WS/DGATa5 | WS/DGATa7 | 5.19 | 1.33 | 3.9 | 0 | 4.26 | |
| WS/DGATa5 | WS/DGATb3 | 5.15 | 1.28 | 4.01 | 0 | 4.12 | |
| WS/DGATa6 | WS/DGATc2 | 1.63 | 3.05 | 0.53 | 0.83 | 1.95 | purifying |
| WS/DGATa6 | WS/DGATb1 | 5.22 | 1.78 | 2.94 | 0 | 4.41 | |
| WS/DGATb1 | WS/DGATb3 | 5.17 | 1.58 | 3.28 | 0 | 4.32 | |
| WS/DGATb1 | WS/DGATb2 | 5.25 | 1.15 | 4.56 | 0 | 4.22 | |
| WS/DGATb2 | WS/DGATc2 | 3.13 | 4.24 | 0.74 | 0.36 | 3.38 | purifying |
| WS/DGATb3 | WS/DGATc2 | 1.78 | 1.53 | 1.16 | 0.44 | 1.72 | |

**Notes.**
Ka, Nonsynonymous substitution rate; Ks, synonymous substitution rate.

at <1, suggesting purifying selection. Our syntenic analysis revealed that the soybean *DGAT* gene family expanded through both segmental and tandem duplications. Furthermore, most orthologous genes are distributed on different chromosomes (Gm09 contains four genes: *DGAT1C*, *DGAT2a3*, *WS/DGAT6*, and *WS/DGAT12*, and Gm13 contains four genes: *DGAT1A*, *cytoDGAT*, *WS/DGAT8*, and *WS/DGAT10*); similarly, most paralogous

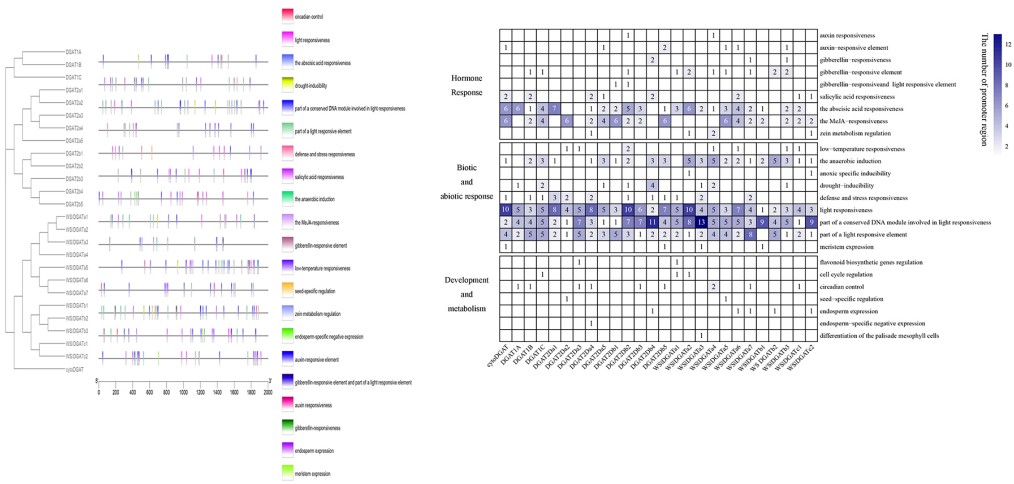

**Figure 3 Promoter analysis.** Promoter structure of soybean *DGAT* genes. (A) Positions of various cis-acting elements in the promoter region 2,000 bp upstream of *DGAT* genes. The black outlined box represents *DGAT* genes with similar promoter structures. (B) The type and frequency of promoter element of *DGAT* genes. The data standardized using absolute value.

genes are distributed on different chromosomes (*WS/DGATa3–WS/DGATa2* on Gm06, *WS/DGATa1–WS/DGATa4* on Gm12, *DGAT2a4–DGAT2a5* on Gm16, *DGAT2b3–DGAT2b4* on Gm16, and *WS/DGATa5–WS/DGATa7* on Gm18), indicating that most duplications of the *DGAT* gene family occur with genome replication.

## Promoter analysis

Positions of promoters of 26 *DGAT* genes are shown in Fig. 3A. Some diverse distribution patterns of cis-acting elements were observed in the promoter region of *GmDGAT* genes, indicating that the DGAT gene family of soybean participates in various biological processes. The analysis of the promoter regions of the 26 *DGAT* genes revealed 25 cis-acting elements that were classified into three divisions (Fig. 3B): hormone response, biotic and abiotic response, and development and metabolism. Light-responsive elements occur in all *DGAT* gene promoters. Elements responsive to abscisic acid and light were found in the promoters of the *DGAT1* subfamily. Except for *DGAT2a1* and *DGAT2b1*, the *DGAT2* subfamily members showed enriched light-responsive and MYB-binding cis-elements in their promoters. Anaerobic induction elements were found in the *WS/DGATa1* gene promoters. Interestingly, although *cytoDGAT* does not share protein domains, it does share promoter features with other *DGAT* genes.

## Spatial DGAT gene expression patterns

The expression patterns of soybean *DGAT* genes in various tissues (root, stem, leaf at different times, flower, seed (stages 1–9), nodule, and shoot) were analyzed from public databases and were found to exhibit evident tissue specificity (Fig. 4). The *DGAT1* subfamily members are primarily expressed during the intermediate stages of grain development; meanwhile, the *DGAT2* subfamily members are primarily expressed in the late stage of grain
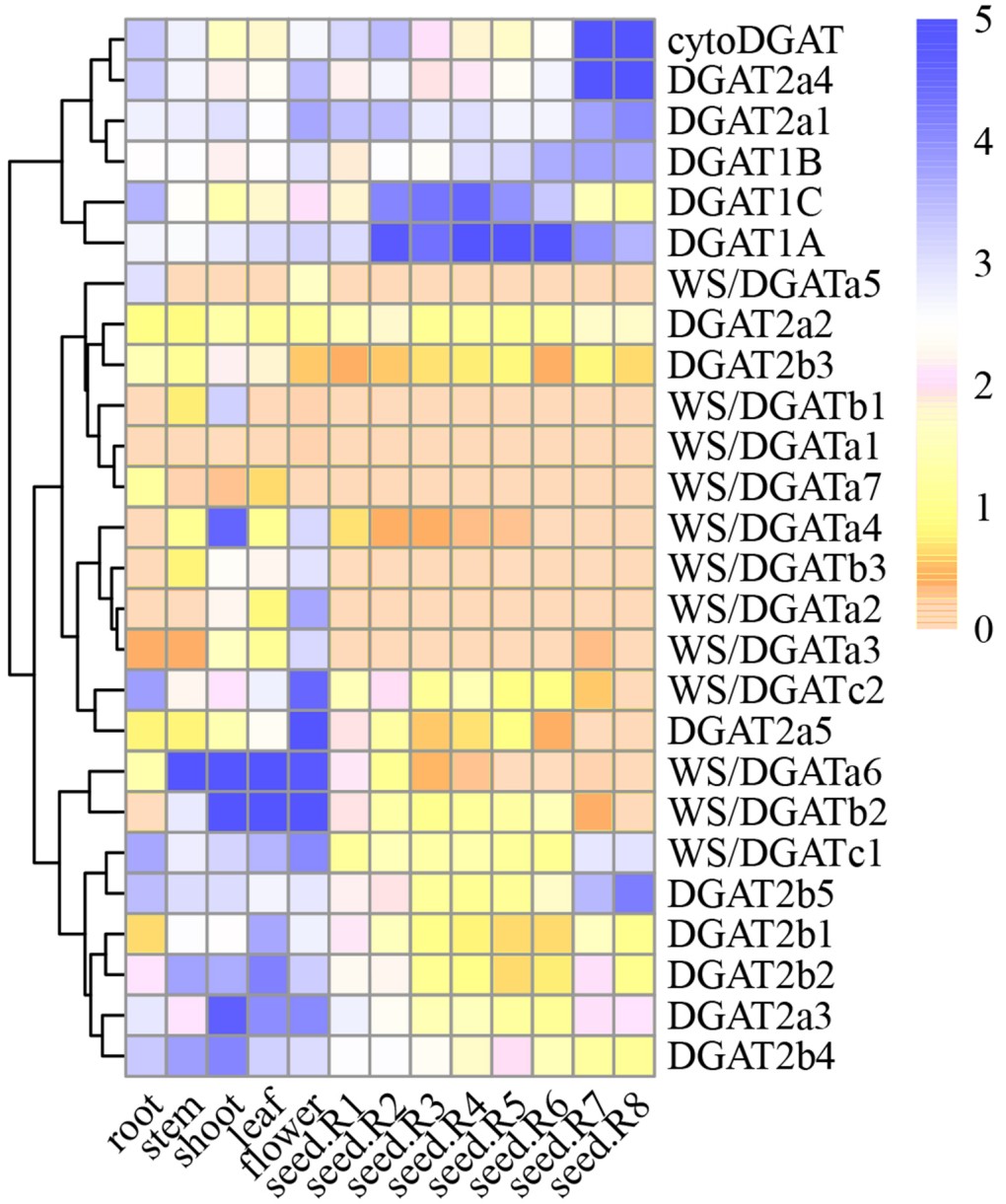

**Figure 4** **GmDGAT expression pattern analysis.** Expression analysis of *GmDGAT* genes in different tissues (roots, stems, leaves, flowers, seeds R-stages 1–8 *i.e.,* reproductive stages from R1 to R8, from flowering to full maturity, and shoots). The data was collected from the data was collected from public repositories, and standardized using a Log2 method. The color depth in the figure represents the amount of gene expression in the sample.

development. *WS/DGAT* subfamily members are expressed in flowers, leaves, and stems as vegetative growth proceeds. In turn, the *cytoDGAT* subfamily members are primarily expressed late during grain development.

In the *DGAT1* gene subfamily, *DGAT1a* showed higher expression than *DGAT1b* or *DGAT1c*, and was expressed in most tissues. The expression of *DGAT1b*, similar to that of

*DGAT1c*, was very low in most tissues, suggesting that *DGAT1a* is the most active gene of the *DGAT1* subfamily. Meanwhile, in the *DGAT2* subfamily, *DGAT2a1*, *DGAT2a3*, and *DGAT2a4* showed high expression levels in the roots, stems, and flowers, respectively. In turn, *DGAT2a1* showed high expression in the early stage of seed formation, whereas *DGAT2a4* showed high expression in the late stage of seed formation. *DGAT2b2*, *DGAT2b4*, and *DGAT2b5* were highly expressed in various tissues. According to the correlation between *DGAT2b1* and *DGAT2b2*, the latter was considered the most important gene among them. As for *DGAT2b3*, *DGAT2b4*, and *DGAT2b5*, the highest level of expression was observed in the mature stage of seed formation; hence, the important gene in this cluster could not be inferred. The expression of *WS/DGATa1*, *WS/DGATa2*, *WS/DGATa3*, *WS/DGATa4*, *WS/DGATa5*, and *DGAT2a7* in various tissues was very low, whereas that of *WS/DGATa6* was very high in stems and leaves.

Considering that a waxy layer covers the outermost surface of the primary aerial plant tissues to reduce water loss, *WS/DGATa6* was inferred to be the important gene in this cluster. *WS/DGATb2* was highly expressed in the stems, leaves, and flowers, and it was the most highly expressed of the three genes in the *WS/DGAT* b cluster. Expression of *WS/DGATc1* in all tissues analyzed was higher than that of *WS/DGATc2*. The 12 genes of the WS/DGAT subfamily were highly expressed in stems, leaves, and flowers, suggesting the involvement of this family in the increased production of wax esters in these tissues. Furthermore, the *cytoDGAT* gene showed a very high level of expression in the mature stage of seed formation and the highest expression level among all gene families during seed formation at stages R7 and R8. Finally, the expression of *WS/DGATA1*, *WS/DGATA2*, and *WS/DGATB1* in various tissues was very low; therefore, we speculate that these may be pseudo or redundant genes.

### Expression of DGAT genes

To further investigate the possible functions of *GmDGATs*, qRT-PCR was used to measure the expression patterns of nine genes highly expressed in subfamilies, in four tissues (root, leaf, stem, and flower), and six stages of soybean embryo (Fig. 5). Nine *GmDGATs* were expressed in the ten tissues, with strong tissue-specific expression patterns. Expression analysis of the *GmDGAT* gene family (Fig. 5) showed that the *GmDGAT1* subfamily members are gradually expressed in maturing pods; in turn, *CtyoDGAT* subfamily members are mainly expressed in the stem meristems, leaves, and mature pods, while *GmDGAT2a1* is expressed in leaves, 10 days after flowering (DAF) pods, and 20 DAF pods. As for *GmDGAT2b1*, this gene is expressed in flowers, stems, and leaves, and the *GmWS-DGAT* subfamily members are expressed in roots, flowers, stems, and leaves.

## DISCUSSION

An accurate analysis of gene family members can contribute to the scientific literature by improving the annotation of genomes. The *GmDGAT* gene family plays important roles in biological processes such as growth and development, seed oil accumulation, and plant stress responses. Additionally, it plays a key role in the accumulation of oil and wax and participates in the anabolism of seed oil and the metabolic process related to stress

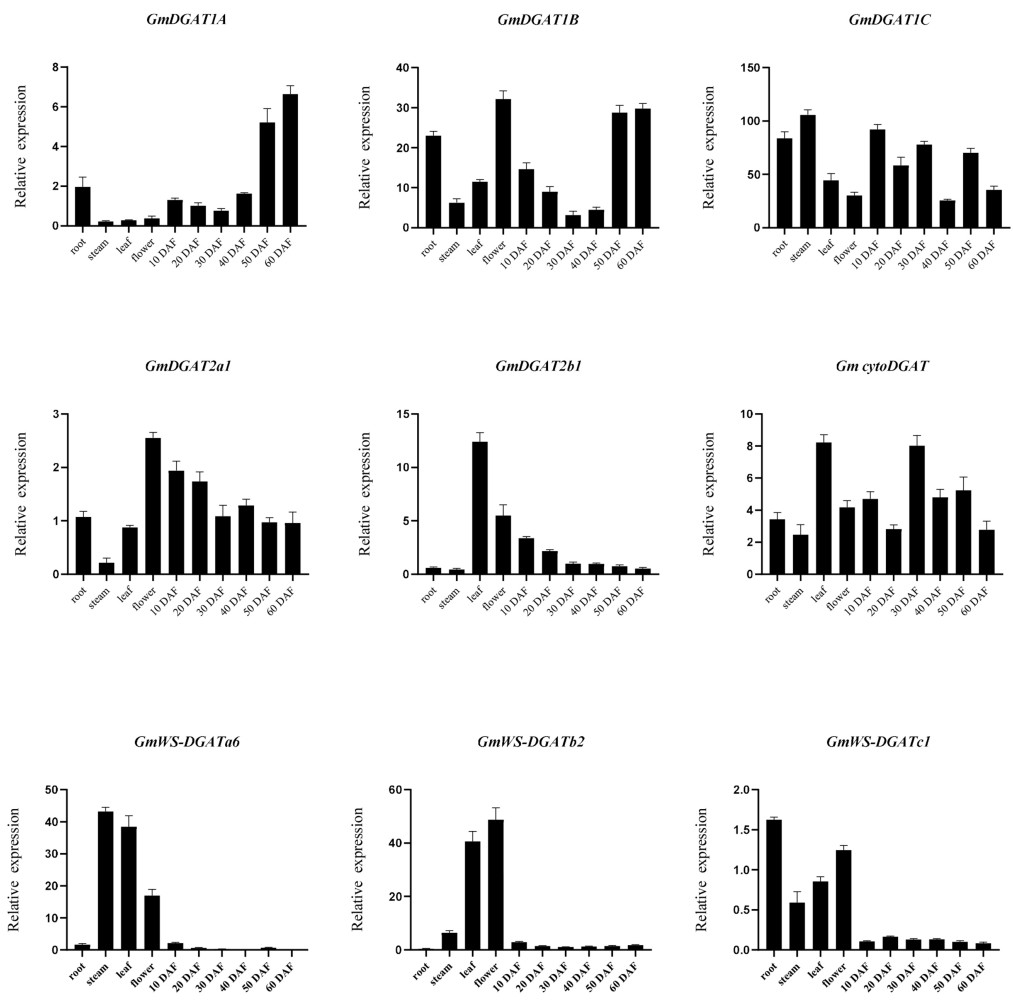

**Figure 5  RT-PCR analysis of GmDGAT genes.** RT-PCR analysis was performed based on the tissue/organ-specific expression pattern of *GmDGAT* (root, stem, leaf, flower, and developing grain at 10, 20, 30, and 40 days after flowering (DAF)). The transcript levels were normalized and expressed relative to the reference *GmAction* gene. Three biological and three technical replications were analyzed using RT-qPCR. Data are the mean ± standard error of three independent replicates.

resistance in plant leaves. However, our research on the *GmDGAT* gene family in soybean mainly focused on the role of the *DGAT1* subfamily in the accumulation of soybean oil. The number of *DGAT2* subfamilies and their relationships remain unclear, with only a few reports on the *WS-DGAT* family and limited research on *cytoDGAT*. In this study, we performed genome-wide detection and characterized 26 putative DGAT proteins within four subfamilies. Further, we identified the number of genes in each subfamily and found differential gene sequences, structures, and conserved variation themes.

GmDGAT2a2, G.max Wm82.a4.v1|Glyma.11G088800.1 CDS has an annotation that does not have a start codon and truncation at the 5′ end of the CDS, relative to other closely related genes. It is truncated by approximately 103 nt from the 5′ end of the CDS.

This missing region of the CDS is present in the genomic sequence, but there is a 1-nt deletion that disrupts the reading frame, suggesting that it is a pseudo gene. GmDGAT2a4, G.max Wm82.a4.v1|Glyma.16G115700.1 CDS has an incorrect annotation. The CDS has an insertion of approximately 30 nt due to an incorrect intron junction in the annotation. The error is highlighted by alignment with two EST sequences (CA953238.1 and BQ299536.1) and by the fact that it has an insertion of approximately 10 amino acids that is not shared by the closely related genes in the gene family.

WS/DGATa1, G.max Wm82. a4. v1 | Glyma.12G114400.1. In this case, the transcription data annotation is incorrect. This CD has 1596 bases, and the transcribed protein with 478 amino acids is different from the transcriptome data of 750 bases and 249 amino acids.

WS/DGATa4, G.max Wm82.a4.v1|Glyma.12G114900.1 has a base deletion at 5′UTR. It may be an error of continuous base sequencing.

Some members of the gene family have unusually large introns >5,000 bp. Thus, for example, GmDGAT1C, G.max Wm82.a4.v1| Glyma.09G065300.1 CDS has a 13,525-bp intron between introns 9 and 10. GmDGAT2a4, G.max Wm82.a4.v1| Glyma.16G115700.1 CDS has a 7971-bp intron between introns 5 and 6. GmDGAT2b1, and G.max Wm82.a4.v1| Glyma.03G243700.1 CDS has a 8480-bp intron between introns 9 and 10.

Arabidopsis contains one *DGAT1* (*AT2G19450*) gene (*Bouvier-Nave et al., 2000*; *Jako et al., 2001*), one *DGAT2* (*AT3G51520*) gene (*Aymé et al., 2014*), one *cytoDGAT* (*At1g48300*) gene (*Aymé et al., 2018*) and no fewer than 11 *WS-DGAT* gene copies (*Li et al., 2017*). According to Rozana's research report (*Rozana et al., 2018*), the oil palm genome contains three, two, two, and two distinctly expressed functional copies of the *DGAT1, DGAT2, DGAT3,* and *WS/DGAT* genes, respectively. Compared with these two plants, the soybean genome contains three, ten, one, and twelve distinctly expressed functional copies of the *DGAT1, DGAT2, DGAT3,* and *WS/DGAT* putative genes, respectively. Although *DGAT1* and *DGAT2* are functionally convergent as they acquired similar types of acyltransferase activity involving diacylglycerol substrates, they are otherwise structurally very distinct from each other. It therefore seems likely that these two enzymes originally evolved separately (*Turchetto-Zolet et al., 2011*). *DGAT3* and *WS/DGAT* have very different evolutionary paths from *DGAT1* and *DGAT2* and seem to originate independently of each other (*Turchetto-Zolet et al., 2016*). As the main oil crop, to improve the oil yield and quality, it is necessary to perform identification and expression analysis of *GmDGAT* gene family in soybean.

The heatmap analysis shows the expression pattern for the *GmDGAT* families within transcriptome libraries, loaded from the public library. Genes with high expression in each family were selected for RT-PCR verification. Expression data and cis-regulatory element prediction revealed the biological function of *GmDGAT* in regulating plant development and resistance. Specific gene expression patterns in different tissues of each subfamily were consistent with the corresponding protein function. Genome-wide identification and analysis of the *GmDGAT* gene family in the soybean (*G. max*) genome provided novel insights into the potential functions of *GmDGAT* genes.

In this study, a total of 26 *DGAT* putative genes were identified, and they were divided into four gene subfamilies (Table 1). Each subfamily showed similar exon and intron
numbers, conserved motifs, and gene structure, but the conserved motifs differed among subfamilies, indicating that the same subfamily plays a similar role in biological functions (Fig. 1). Chromosome localization and collinearity analysis showed that the generation of gene subfamilies was associated with chromosome replication (Fig. 3), which may be due to whole genome duplication during evolution (*Schlueter et al., 2007*). In the process of biological evolution, gene replication events and functional differentiation are relatively important forces driving genome and species evolution. The results of selection pressure show that most *DGAT* genes have resulted from positive selection pressure (Table 2) and have been relatively conserved through evolution, most likely owing to the need to maintain their functional stability as enzyme proteins. Promoter analysis (Fig. 3) showed that subfamilies *DGAT1* and *DGAT2* contained light-responsive elements, which was related to their participation in seed oil accumulation (*Chattopadhyay et al., 1998*). In turn, *WS-DGAT* contained anaerobic-inducing elements, which are related to the formation of stress-related plant lipid membranes. The expression of *GmDGAT* genes showed clear tissue specificity. Further, the *DGAT1* subfamily was mainly expressed in the middle stage of grain development (Figs. 4 and 5), and the *DGAT2* subfamily was mainly expressed in late grain development, with the progress of vegetative growth, while *WS/DGAT* was expressed in flowers, leaves, and stems. The expression of each subfamily member in different tissues also reflects its corresponding biological functions in plant growth and development. Such a large number of *DGAT*-related genes may be due to their functional redundancy and functional differentiation in soybean during evolution to adapt to the environment, thus showing functional diversity. This phenomenon is also related to the characteristic chromosome- and genome-doubling of soybean. Our qPCR results verified the specific expression profile, and the highly expressed genes in each subfamily were closely related to their biological functions.

## CONCLUSIONS

Here, for the first time, we comprehensively analyzed the soybean *DGAT* gene family members. In total, 26 *DGAT* genes were identified and found to be unevenly distributed across the soybean genome. Furthermore, these genes were categorized into four groups through phylogenetic analysis, the results of which were supported by exon–intron structure, motif composition, and conserved domain analysis. We identified 25 cis-elements in the *DGAT* promoter regions, all of which were associated with responses to plant hormones and light, resistance, seed development, and common elements. The Ka/Ks ratio of most *GmDGAT* genes was estimated as >1, indicating that they are under significant positive selection pressure. Gene expression analysis revealed that the *DGAT* family exhibits tissue-specific characteristic expression patterns, with *DGAT2a4* and *cytoDGAT* being primarily expressed in the late stage of seed maturity and probably involved in oil accumulation in late-maturing seeds. Meanwhile, *WS/DGATa6* expression in the stems and leaves was very high, probably due to the epidermal waxy layer. These

findings provide important information for research on the soybean fatty acid metabolism pathways.

### Funding

This research was supported by the Major Science and Technology Sponsored Program for Transgenic Biological Breeding (grant No. 2016ZX08004-003); by the National Natural Science Foundation of China (grant No. 3210150556); and by the Science and Technology Project of the 13th five-year Plan of Jilin Province Education Department (grant No. JJKH20201020KJ). The funders had no role in study design, data collection and analysis, decision to publish, or preparation of the manuscript.

### Grant Disclosures

The following grant information was disclosed by the authors:
Major Science and Technology Sponsored Program for Transgenic Biological Breeding: 2016ZX08004-003.
National Natural Science Foundation of China: 3210150556.
Science and Technology Project of the 13th five-year Plan of Jilin Province Education Department: JJKH20201020KJ.

### Competing Interests

The authors declare there are no competing interests.

### Author Contributions

- Shihui Zhao performed the experiments, prepared figures and/or tables, authored or reviewed drafts of the article, and approved the final draft.
- Fan Yan performed the experiments, prepared figures and/or tables, authored or reviewed drafts of the article, and approved the final draft.
- Yajing Liu analyzed the data, prepared figures and/or tables, authored or reviewed drafts of the article, and approved the final draft.
- Monan Sun analyzed the data, prepared figures and/or tables, authored or reviewed drafts of the article, and approved the final draft.
- Ying Wang analyzed the data, prepared figures and/or tables, authored or reviewed drafts of the article, and approved the final draft.
- Jingwen Li analyzed the data, prepared figures and/or tables, authored or reviewed drafts of the article, and approved the final draft.
- Xinsheng Zhang analyzed the data, prepared figures and/or tables, authored or reviewed drafts of the article, and approved the final draft.
- Xuguang Yang conceived and designed the experiments, prepared figures and/or tables, authored or reviewed drafts of the article, and approved the final draft.
- Qingyu Wang conceived and designed the experiments, prepared figures and/or tables, authored or reviewed drafts of the article, and approved the final draft.

## Data Availability

The raw data is available in the Supplemental Files.

## Supplemental Information

Supplemental information for this article can be found online at http://dx.doi.org/10.7717/peerj.14941#supplemental-information.

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
