# Peer review of "Genome-wide identification and expression analysis of diacylglycerol acyltransferase genes in soybean (*Glycine max*)"

_PeerJ, doi:10.7717/peerj.14941_

## Round 0.1 · original submission · Major Revisions

Please address concerns raised by the reviewers and amend manuscript accordingly.

·

Basic reporting

Many places with typos and grammatical errors. They have been pointed out and suitable edits have been suggested. Some figures need restructuring and clarification. There are major comments that need to be addressed with necessary additions and edits in the main text. Please see the attached reviewer's comment for the details.

Experimental design

Some methods subsections and many places in the results section need further explanations. Please see the attached reviewer's comment for the details.

Validity of the findings

Some raw data are available. I have asked for further raw data showing the calculations/ analysis. Please see the attached reviewer's comment for the details.

Additional comments

Please check the overall language, grammar, and structural organization of the manuscript thoroughly. Details are in the attached reviewer's comment for the details.

Reviewer 2 ·

Basic reporting

The manuscript by Zhao et al, was trying to analyze the diacylglycerol acyltransferase genes (DGAT) family in soybean (Glycine max). While the topic could be interesting, there are many weaknesses that they may need to address in the paper.

Major
1, The comparison of DGAT protein family genes with these in other model plants, such as Arabidopsis and rice is not present. For example, the authors may refer to some relevant papers when performing the analysis (https://www.nature.com/articles/s41598-020-66860-7).

2, For the result in Figure 1a, the author should descript what is in the 4th panel.

3, For the result in Figure 3, what does the color indicate? The number from 1 to 12, what is that?

4, For figure 4, where does the data come from? Are they from a public database? If so they should describe this in the Materials and Methods.

5, For figure 5 where the author examined the expression using RT-PCR, how samples are prepared, such as the growth condition of plants (temperature, light intensity, soil/medium), which leaves were used, how to sample roots … They should describe this in very detail in the Materials and Methods.

Experimental design

4, For figure 4, where does the data come from? Are they from a public database? If so they should describe this in the Materials and Methods.

5, For figure 5 where the author examined the expression using RT-PCR, how samples are prepared, such as the growth condition of plants (temperature, light intensity, soil/medium), which leaves were used, how to sample roots … They should describe this in very detail in the Materials and Methods.

Validity of the findings

no

Additional comments

The result and discussion are too short for a research paper. They should write this very detail so that readers can follow what are the findings.

Reviewer 3 ·

Basic reporting

I have two major points.
1. Showing gene sequences does not always mean that the sequence codes for a functional enzyme. The authors should use the term 'putative' throughout their manuscript, including the title.
2. The cited references to DGAT are unusual. There has been quite a lot of work with major oil crops (including soybean) which are not referred to and should be. A useful major review on DGAT which has just been published is that by Chen, G, et al. 2022 Progress in Lipid Research 88, 101181 and this could be used by the authors to improve their manuscript.

Experimental design

No problems

Validity of the findings

See my comment above about using the term 'putative gene' throughout.

Additional comments

These specific comments are for the authors to improve their manuscript.
line 16. The term 'rate-limiting' is often misleading. It depends on the conditions. The phrase ' can exert strong flux control' is much better.
22. Define Ka/Ks.
34-35. Use the terms 'triacylglycerols' and 'non-esterified fatty acids'
39 and throughout. Rephrase 'rate-limiting' throughout (see above remark)
41. Is fatty acyl-CoAs meant?
150. Give detail for how the isoelectric point was determined.
207. Define/explain 'syntenic'.
258-305. The gene expression patterns are simplified and do not always correspond to what is written in the text. They should be revised. In addition, the data in Figures 4 and 5 should be compared.

---

## Round 0.2 · Minor Revisions

Please address the remaining concerns of the reviewers and amend the manuscript accordingly.

·

Basic reporting

The authors have satisfactorily addressed most of my concerns in the revised form.
Regarding Fig 1ai, the proteins scale at the bottom should be labeled as N- and C terminal; not 5' and 3' as in the current form (which is for nucleotides, not proteins).

Experimental design

No additional comments.

Validity of the findings

No additional comments.

Reviewer 2 ·

Basic reporting

I am satisfied with the responses from the authors. I have no other questions.

Experimental design

none

Validity of the findings

none

Additional comments

none

Reviewer 3 ·

Basic reporting

In the revised submission, the authors have dealt with most of the queries raised in a satisfactory manner. The manuscript is just about acceptable now.

The editors might like to note that reviewer 1 found a number of discrepancies in the original data which have now been changed. It is a pity that the authors were not more careful in the first place!

Experimental design

O,k, now that the m/s has been revised.

Validity of the findings

o.k.

---

## Round 0.3 · accepted · Accept

All remaining concerns of the reviewers were adequately addressed and therefore revised version is acceptable now.